# Dual-Targeted Extracellular Vesicles to Facilitate Combined Therapies for Neuroendocrine Cancer Treatment

**DOI:** 10.3390/pharmaceutics12111079

**Published:** 2020-11-11

**Authors:** Yingnan Si, JiaShiung Guan, Yuanxin Xu, Kai Chen, Seulhee Kim, Lufang Zhou, Renata Jaskula-Sztul, X. Margaret Liu

**Affiliations:** 1Department of Biomedical Engineering, University of Alabama at Birmingham (UAB), 1825 University Blvd, Birmingham, AL 35294, USA; yingnan@uab.edu (Y.S.); guan0926@uab.edu (J.G.); yuanxin8@uab.edu (Y.X.); kaisdzb@uab.edu (K.C.); seulheekim@uabmc.edu (S.K.); 2Department of Medicine, University of Alabama at Birmingham, 703 19th Street South, Birmingham, AL 35294, USA; lfzhou@uab.edu; 3Department of Surgery, University of Alabama at Birmingham, 1808 7th Avenue South, Birmingham, AL 35294, USA; rjsztul@uabmc.edu

**Keywords:** dual targeting delivery, mAb-EV, combined chemotherapies, neuroendocrine cancer

## Abstract

Neuroendocrine (NE) cancers arise from cells within the neuroendocrine system. Chemotherapies and endoradiotherapy have been developed, but their clinical efficacy is limited. The objective of this study was to develop a dual-targeted extracellular vesicles (EV)-delivered combined therapies to treat NE cancer. Specifically, we produced EV in stirred-tank bioreactors and surface tagged both anti-somatostatin receptor 2 (SSTR 2) monoclonal antibody (mAb) and anti-C-X-C motif chemokine receptor 4 (CXCR4) mAb to generate mAbs-EV. Both live-cell confocal microscopy imaging and In Vivo Imaging System (IVIS) imaging confirmed that mAbs-EV specifically targeted and accumulated in NE cancer cells and NE tumor xenografts. Then the highly potent natural cytotoxic marine compound verrucarin A (Ver-A) with IC_50_ of 2.2–2.8 nM and microtubule polymerization inhibitor mertansine (DM1) with IC_50_ of 3.1–4.2 nM were packed into mAbs-EV. The in vivo maximum tolerated dose study performed in non-tumor-bearing mice indicated minimal systemic toxicity of mAbs-EV-Ver-A/DM1. Finally, the in vivo anticancer efficacy study demonstrated that the SSTR2/CXCR4 dual-targeted EV-Ver-A/DM1 is more effective to inhibit NE tumor growth than the single targeting and single drug. The results from this study could expand the application of EV to targeting deliver the combined potent chemotherapies for cancer treatment.

## 1. Introduction

Neuroendocrine (NE) cancers arising from cells within the neuroendocrine system are often metastatic at the time of initial diagnosis [1], and patients with untreated, isolated NE tumor (NET) liver metastases have a <30% five-year survival rate. Several chemotherapies, such as everolimus, sunitinib, resveratrol, octreotide, thailandepsin-A and AB3 [2,3,4,5,6,7,8,9,10,11,12,13,14,15], have been investigated for NET treatment, but their therapeutic efficacy is limited. The peptide receptor radionuclide therapy (PRRT, Lutathera) that combines endoradiotherapy (^177^Lu-DOTA-TATE) has been approved to treat somatostatin receptor (SSTR) positive gastroenteropancreatic NETs [16,17], but it has short radiopharmaceutical shelf life, reduces active concentration over time due to the decay of ^177^Lu, and has a poor therapeutic impact on rapidly proliferating NE cancers. There is no effective treatment for metastatic NETs.

Extracellular vesicles (EV) play important roles in cell–cell communication via transporting miRNA, siRNA, DNA, and proteins in vivo, and have great potential to deliver drug for cancer treatment and to monitor cancer progression [18,19,20,21,22,23]. In a previous study, we developed a stirred-tank bioreactor-facilitated EV biomanufacturing platform, established monoclonal antibody (mAb) surface tagging technology, and validated the targeting delivery efficacy of mAb-EV [24]. Despite this achievement, targeting a single receptor has the limitation of treating the heterogeneous cancers and possible antigen loss could significantly reduce the cancer treatment efficacy. Moreover, cancer cells often develop drug resistance and recurrence after single therapy treatment. Therefore, it is very important to develop a dual-targeted vehicle to deliver combined therapies with synergistic anticancer mechanisms.

The SSTR2 has been used as a target [25,26] in NET diagnosis [27,28,29] and treatment via SSTR analog octreotide, ^90^Y or ^177^Lu-PRRT [2,3,4,5,16,17]. Literature [30,31,32], clinical [33,34,35] and our tissue microarray analysis (TMA) [36] showed that ~75% samples had high SSTR2 expression, including >70% low-grade (G1, G2) NETs [37] and >20% high-grade (G3) NETs [38] (*n* = 38–151). Clinical data revealed that anti-C-X-C motif chemokine receptor 4 (CXCR4) is crucial for metastases of highly proliferative (G3) NETs [37,38] accounting for 12–22% patients (*n* = 137–164). Literature [37,38,39,40] demonstrated that CXCR4 is positive in ~50% G2 and ~80% G3 NETs and a good target to diagnose and treat highly proliferative NETs [37,41,42]. It is estimated that dually targeting SSTR2/CXCR4 to treat SSTR2^+^ and/or CXCR4^+^ NET patients could cover 95% primary and metastatic NETs.

To specifically target NETs, we have developed a novel anti-SSTR2 mAb (IgG1, κappa) [24,36] which targets the first and second extracellular domains of SSTR2 [43]. Our mAb exhibited high specificity, strong binding and effective drug delivery to SSTR2^+^ NET as confirmed in immunohistochemistry of NET cell lines, tissues and xenograft model. The anti-CXCR4 mAbs (e.g., Ulocuplumab [44], PF-06747143 [45]) are being evaluated in Phase I/II clinical trials for relapsed myeloma and chronic lymphocytic leukemia treatment. This study will construct anti-SSTR2/CXCR4 mAbs-EV as a dual-targeted drug delivery vehicle for NET treatment.

Marine natural compounds are a significant source of novel anticancer leads [6,46]. Verrucarin A (Ver-A), isolated from the salt water culture of *Myrothecium verrucaria* [47], shows strong antiproliferative and proapoptotic responses in several cancer cell lines including renal carcinoma [48], hepatocellular carcinoma [49], and breast carcinoma [50]. The mertansine (DM1) is a highly potent cytotoxic agent that inhibits the polymerization of microtubules during cell proliferation [51]. DM1 has been applied to treat cancer as a part of antibody–drug conjugates such as the U.S. Food and Drug Administration (U.S. FDA)-approved trastuzumab emtansine, but the free DM1 is too toxic to be used a chemotherapy. We have found that both Ver-A and DM1 are potent growth inhibitors of NET cells at low nanomolar concentrations. Therefore, we hypothesize that the combination of Ver-A and DM1 could integrate two anticancer mechanisms and effectively treat NE cancers.

The objective of this study was to develop and validate a dual-targeted EV to deliver combined potent chemotherapeutic drugs. Our anti-SSTR2 mAb and a commercial anti-CXCR4 mAb were tagged to EV to generate dual-targeted mAbs-EV to deliver Ver-A and DM1. The in vitro and in vivo evaluation of SSTR2/CXCR4 mAbs-EV-Ver-A/DM1 showed high NET-specific targeting and high efficacy to inhibit tumor growth in xenograft mouse model. This study demonstrated the feasibility to use EV for dual receptors targeting and combined therapies delivery, which would expand the application of EV in cancer treatment.

## 2. Materials and Methods

The animal studies conform to the Guide for the Care and Use of Laboratory Animals published by the National Institutes of Health (NIH Publication No. 85-23) and have been approved by the Institutional Biosafety Committee at the University of Alabama at Birmingham under the animal project number of IACUC-21929 (approved on 12 December 2019).

### 2.1. Cell Lines, Seed Cultures and Media

The metastatic pancreatic NET cell lines BON-1 (kindly provided by Dr. Mark Hellmich at University of Texas, USA) and QGP-1 (AcceGen Biotechnology, Fairfield, NJ, USA), BON-FLuc generated by stably transfecting BON cells with luciferase-expressing plasmid pGL4.50[luc2/CMV/Hygro (Promega, Madison, WI, USA) using lipofectamine 2000 (Life Technologies, Part of Fisher, Carlsbad, CA, USA), thyroid NET cell lines MZ-CRC-1 (provided by Dr. Alexander Knuth, Department of Oncology Krankenhaus Nordwest, Frankfurt, Germany) and TT (ATCC, Manassas, VA, USA), and pulmonary NET cell line H727 (ATCC) were used for in vitro or in vivo studies. BON, QGP, and MZ cells were maintained in Dulbecco’s Modified Eagle Medium: Nutrient Mixture F-12 (DMEM/F12) with 10% fetal bovine serum (FBS) and 4 mM l-glutamine, and TT cells (ATCC) were maintained in Roswell Park Memorial Institute 1640 medium (RPMI-1640) with 20% FBS in T25 or T75 flasks. The noncancerous foreskin fibroblast cell line 917 (donated by Dr. Joseph Goldstein from University of Texas Southwestern Medical Center, USA) was maintained in DMEM with 10% FBS, 1% nonessential amino acids, and 1% sodium pyruvate at 37 °C and 5% CO_2_ in a humidified incubator (Caron, Marietta, OH, USA). The seed train of anti-SSTR2 mAb producing hybridoma was maintained in hybridoma serum free medium with 8 mM L-glutamine in 125-mL shaker flasks at 37 °C, 5% CO_2_ and 130 rpm. The human embryonic kidney 293 (HEK 293F) cells (Life Technologies) were maintained in chemically defined HEK 293 medium with 4 mM GlutaMAX in 125 or 250-mL shaker flasks. The viable cell density (VCD) and viability was measured using a Countess II automated cell counter (Fisher Scientific, Waltham, MA, USA). All basal media and nutrient supplements were purchased from Thermo Fisher Scientific (Waltham, MA, USA) or Life Technologies unless otherwise specified.

### 2.2. Construction of mAbs-EV-Ver-A/DM1

First, the EV were produced using our previously published platform [24]. Briefly, 2 L stirred-tank bioreactors (Vineland, NJ, USA) were utilized to conduct EV production. The suspension HEK 293F cells were seeded into the bioreactor vessel with 1.5-L FreeStyle^TM^ 293 expression medium (GlutaMAX) at seeding density of 0.3–0.5 × 10^6^ c/mL. A total of 4 g/L Cell Boost 6 was added to basal medium and glucose was supplied to maintain a range of 2–6 g/L during EV production. The production was performed with parameters controlled at Temp 37 °C, pH 7.0, DO 50% and agitation 70 rpm. The EV were purified following our previously reported procedure [24]. Briefly, when HEK293F cell viability dropped to 80–90% in the bioreactor, the culture broth was collected and centrifuged at 4000 *g* and 4 °C for 1 h. Then the collected spent medium was filtered to remove cell debris and treated with proteinase K to remove secreted host cell proteins. The EV were purified, concentrated and washed using Vivaspin 300 kDa MWCO concentrators (Fisher Scientific, Waltham, MA, USA). Finally, EV were further purified and washed using Vivaspin turbo 100 kDa MWCO concentrators (Fisher Scientific, Waltham, MA, USA). In this study, the fine polishing using Protein A chromatography was not performed. The purified EV were used for surface tagging and drug packing or supplemented with 25 mM trehalose for long-term storage at −80 °C. Second, following the reported procedure [24,36], the anti-SSTR2 mAb was produced by hybridoma cells in 2-L bioreactors and purified using New Generation Chromatography (NGC) system (Bio-Rad, Hercules, CA, USA) equipped with Protein A column. The anti-SSTR2 mAb was titrated using high performance liquid chromatography (HPLC) system (Shimadzu, Columbia, MD, USA) equipped with a UV detector and a viva C-8 column (Phenomenex, Torrance, CA, USA). Third, both anti-SSTR2 mAb and anti-CXCR4 mAb were tagged to EV via 1,2-Distearoyl-*sn*-Glycero-3-Phosphoethanolamine polyethylene glycol succinimidyl ester (DSPE-PEG-NHS) linker to generate mAbs-EV, with mPEG-DSPE stabilizer (New York, NY, USA) to improve the circulation stability, as we reported previously [24]. The generated mAbs-EV were titrated with NanoSight (Malvern Panalytical, Malvern, UK) with five captures in 5 min per sample using SCMOS camera at room temperature, pump speed of 25 and detection threshold of 5. The morphology of EV was confirmed with a Tecnai Transmission electron microscope (TEM, FEI, Hillsboro, OR, USA).

### 2.3. Packing Ver-A, DM1 and Cyanine 7 (Cy7) in mAbs-EV

Total of 10 × 10^10^ EV were incubated with 0.116 mg (214 nanomole) of Ver-A and 0.25 mg of DM1 in 8.6 mL of PBS for overnight at room temperature. The unpacked free drugs were removed using Vivaspin 20 300 kDa column (Fisher Scientific, Waltham, MA, USA) and HPLC (Shimadzu, Kyoto, Japan) equipped with column of 250 × 4.6 mm^2^ 5 micron, 5 u C18 300A (Phenomenex) was used to titrate Ver-A and DM1 with setup of Buffer A-H_2_O + 0.1% formic acid, Buffer B-acetonitrile + 0.1% formic acid, and flow rate of 1 mL/minute to analyze the packing rate. The Cy7-PE or Liss Rhod-PE fluorescent dye (Alabaster, Alabama, AL, USA) (16.7 nmol) was used to label 1 × 10^12^ EV via 1,2-dioleoyl-*sn*-glycero-3-phosphoethanolamine-N (Fisher Scientific, Waltham, MA, USA) in the dark at room temperature with horizontal shaking overnight. The fluorescence labelled mAbs-EV or mAbs-EV-drugs can be used to monitor the in vitro uptake and in vivo biodistribution/cancer-specific targeting.

### 2.4. Flow Cytometry

Flow cytometry analysis was performed to quantify the surface expression of SSTR2 and CXCR4 receptors in multiple NET cell lines (BON, QGP, H727, TT and MZ) and a negative control fibroblast cell line (917) using a flow cytometer (LSRII, BD Biosciences, San Jose, CA, USA). The anti-SSTR2 mAb or anti-CXCR4 mAb (RD Systems, Minneapolis, MN, USA) was labelled with an Alexa Fluor™ 647 labelling kit to generate fluorescent AF647-mAb. The staining conditions are 1 µg AF647-mAb per million cells at 4 °C for 30 min as reported before [24,36,51,52].

### 2.5. Confocal Imaging

The in vitro uptake of mAbs-EV by NET cell lines was confirmed with two- or three-color confocal microscopy [24,36,51,52]. The cytoplasm and nucleus were stained with BacMam GFP Transduction Control (Green); the nucleus was stained with DAPI (blue); and the Liss Rhod fluorescent dye (red) was used to monitor the uptake of mAbs-EV. An Olympus 1X-81 confocal microscope with a laser scanning head (Olympus IX81, Center Valley, PA, USA) was used to scan the stained NET cells and collect MitoSox images.

### 2.6. Western Blotting

The NuPAGE™ 4–12% Bis-Tris protein gels were used to run nonreducing SDS-PAGE. The primary mouse antihuman antibody and horseradish peroxidase (HRP)-conjugated secondary anti-mouse antibody (Abcam, Cambridge, UK) were used to detect the intracellular biomarkers and followed with treatment using Luminata Forte Western HRP substrate (Millipore, Boston, MA, USA). The blotted membrane was imaged with MyECL imager (Thermo Scientific, Waltham, MA, USA) and quantified with ImageJ software (National Institutes of Health, Bethesda, MD, USA).

### 2.7. In Vivo Imaging System (IVIS) Imaging

The six-week-old nude (nu/nu) mice (mus) (Jackson Labs, Sacramento, CA) were subcutaneously (s.c.) injected with 3 × 10^6^ BON-FLuc cells or QGP cells into the left flank. When tumor volumes reached ~75–100 mm^3^, the mAbs-EV-Cy7 or mAbs-EV-Ver-A/DM1 were administered via the tail vein for targeting analysis or anticancer treatment, respectively. The in vivo NET targeting and biodistribution of Cy7 labeled mAbs-EV was evaluated using IVIS. At 24 h after i.v. injections of 30 × 10^10^ mAbs-EV-Cy7, IVIS imaging was performed to capture the Fluc bioluminescence (tumor) and Cy7 fluorescence (mAbs-EV). The NET-specific targeting was confirmed by the colocalization of bioluminescence and fluorescence.

### 2.8. In Vitro Anticancer Cytotoxicity

The in vitro anticancer cytotoxicity assay was performed following our published protocol [53]. Briefly, the BON and QGP cells were seeded in 96-well plates with seeding density of 1 × 10^5^ cells/mL, viability of >95%, and volume of 100 µL/well. The Ver-A and DM1 dosages of 0.5, 1, 5, 10, 20, and 40 nM were tested to calculate the IC_50_ values. The CellTiter-Glo Luminescent Cell Viability Assay (Promega) was used to measure the cell viability after three days’ treatment.

### 2.9. Maximum Tolerated Dose (MTD)

To investigate the tolerated dosages and potential toxicity, five different doses of EV-Ver-A/DM1 (molar ratio of Ver-A:DM1 is 1:1), i.e., 0.31, 0.62, 1.24, 2.48 and 4.96 mg/kg, were administrated into non-tumor-bearing nude mice via intraperitoneal (i.p.) injection (*n* = 3). Mice were monitored twice daily for a total of eight days and showed no overt changes in general health including water intake, breathing, and locomotion. At the end of the study, mice were sacrificed to collect the major organs including brain, lung, heart, kidney and liver for Hematoxylin and Eosin (H&E) staining to analyze the potential toxicity of Ver-A/DM1.

### 2.10. In Vivo Anticancer Efficacy

The NET (QGP and BON-FLuc) xenografted mice with established tumor were randomized into four groups (*n* = 6) and treated with PBS, SSTR2 mAb-EV-DM1 (1.24 mg/kg), SSTR2 mAb-EV-Ver-A/DM1 (1.24 mg/kg), and SSTR2/CXCR4 mAbs-EV-Ver-A/DM1 (1.24 mg/kg) in 100 µL of PBS. Two injections were carried out on days 5 and 10. Throughout the three-week treatment, tumor size was measured every other day.

### 2.11. Hematoxylin and Eosin (H&E) Staining

The tissue samples were embedded in paraffin, sectioned at 5 μm with Leica microtome and mounted on frosted microscope slides at University Pathology Facility. During H&E staining, the embedded slides were dewaxed with xylene and gradient hydrated with 100–0% ETOH, then immersed in hematoxylin solution for 10 min followed by tap water rinse for 5 min and eosin rinse for 12 dips. The slides were dipped in ddH_2_O until the eosin stopped streaking, in 50% ethanol 10 times, and in 70% ethanol 10 times. Then, the slides were equilibrated in 95% ethanol for 30 s and 100% ethanol for 1 min. Finally, the slides were dipped in xylene five times and mounted with cytoseal Xyl.

### 2.12. Statistical Analysis

All the experimental data were presented as mean ± standard error of the mean (SEM). Two-tailed Student’s *t* tests were used to determine the probability of significance between groups. Comparison was performed using a one-way ANOVA followed by post hoc (Dunnett’s) analysis. Statistical significance with ** *p* value of <0.005 was considered for all tests.

## 3. Results and Discussion

### 3.1. SSTR2 and CXCR4 Surface Expression and Dual Targeting

The surface expression of SSTR2 and CXCR4 in NET cell lines was evaluated by staining BON, QGP, MZ and TT cells with SSTR2 and CXCR4 antibodies at 4 °C. As presented in Figure 1, flow cytometry analysis showed that the metastatic pancreatic NET QGP cell had high SSTR2 and CXCR4 expression with binding rate of 65% and 71%, respectively. The metastatic pancreatic NET BON cells had high-level SSTR2 expression (66%) while medium-level CXCR4 expression (47%). The metastatic medullary thyroid NET MZ cells had medium-level surface SSTR2 (40%) and high-level surface CXCR4 (77%). The expression levels of both receptors were relatively low (25%) in medullary thyroid NET TT cells. The pulmonary NET H727 cells had very low SSTR2 (9%) and CXCR4 (10%) expression. The expression level of SSTR2 and CXCR4 in noncancerous foreskin fibroblast cell line 917 was 6% and 3%, respectively.

SSTR2 is considered as a good target for low-grade (G1/G2) of NETs. Our previous immunohistochemistry staining of G1 and G2 NET patient tissues (*n* = 37) showed that 76% samples had high SSTR2 expression [36]. Leijon et al. also reported 74.8% of 151 NET tissues, including 13/14 metastasized samples, overexpressed SSTR2 [33]. We expect that the anti-SSTR2 mAb-EV-drug can target and treat ~75% SSTR2^+^ NET patients which are comprised of most G1/2 and >20% G3. Furthermore, CXCR4 plays an important role in highly proliferative (G3b) NETs [37,38] and literature reported that CXCR4 is positive in 50% G2 NETs and 80% G3 NETs (*n* = 137–164) [37,38,39,40]. All these data indicated that SSTR2/CXCR4 dual-targeted EV-drugs could cover ~95% patients with differentiated and metastatic NETs, either SSTR2^+^ or CXCR4^+^. Moreover, dual targeting could also benefit the patients with highly heterogeneous NET. For the SSTR2^−^/CXCR4^−^ NET patients, we can consider targeting an alternative receptor, such as the membrane gastric inhibitory polypeptide receptor (GIPR), which is overexpressed in both primary (*n* = 103) and metastatic (*n* = 123) NET tissues [54].

The bispecific antibodies have been developed to achieve dual targeting by combining the two antigens binding fragments of monoclonal antibodies into one antibody. However, the needed antibody engineering is time-consuming and has low flexibility to combine different antibodies. As compared to a bispecific antibody, our dual-targeted mAbs-EV is flexible to target different combination of multiple surface receptors in cancer cells.

### 3.2. NET-Specific Targeting and Uptake of Dual-Targeted mAbs-EV

The homogeneity, quality and size distribution of mAbs-EV were characterized. The transmission electron microscope (TEM) image showed that the targeted EV were physically homogenous (Figure 2A). The nanoparticle tracking analysis using NanoSight showed that the mAbs-EV had a mean size of 145 ± 39 nm with size distribution of 102–194 nm (Figure 2B). Moreover, our previous study confirmed the endosomal origin of the generated EV and high purity without contamination of ER, Golgi, mitochondria and nuclear [24]. The live-cell confocal laser scanning microscopy (CLSM) imaging was performed to assess the targeting delivery capability of anti-SSTR2/CXCR4 mAbs-EV. As shown in Figure 2C, the mAbs-EV-Liss Rhod (displayed as red color) targeted and internalized into the cytoplasm of BON and QGP cells. These results indicated that mAbs-EV can bind NET cells and deliver drugs intracellularly.

Furthermore, we evaluated the in vivo NET-targeting specificity of Cy7-labeled mAbs-EV, using BON-FLuc and QGP s.c. xenografted nude mouse model. The live-animal IVIS imaging at 24 h after mAbs-EV-Cy7 injection showed that the BON-FLuc xenograft bioluminescence signal or QGP xenograft overlapped with the Cy7 fluorescence signal, suggesting that the anti-SSTR2/CXCR4 mAbs-EV preferentially bound to NE tumors (Figure 2D). It is also found that the accumulation of mAbs-EV in QGP xenograft was higher than that in BON xenograft, which was probably caused by the higher expression of SSTR2 and CXCR4 receptors in QGP. Moreover, the ex vivo IVIS analysis of tumor and main organs (heart, lung, brain, pancreas, spleen and liver) confirmed the NET xenograft-specific trageting of dual-targeted mAbs-EV and lack of binding to major organs (Figure 2E). Altogether, these data proved that the constructed SSTR2/CXCR4 mAbs-EV can specifically target NET cells both in vitro and in vivo.

### 3.3. In Vitro Anticancer Cytotoxicity and Synergistic Mechanisms

The in vitro anticancer cytotoxicity of the Ver-A and DM1 was tested in a three-day assay (Figure 3A). Multiple dosages of these drugs, including 0.5, 1, 5, 10, 20, and 40 nM, were evaluated using both BON and QGP cells. As described in Figure 3A, Ver-A showed the highest toxicity in BON cells, which killed >90% cells at >10 nM within three days. The calculated IC_50_ values of Ver-A were 2.8 nM for BON cells and 2.2 nM for QGP cells, and the IC_50_ values of DM1 were 4.5 nM for BON cells and 3.2 nM for QGP cells. These results indicated that the Ver-A and DM1 led to the growth reduction at low, single-digit nanomolar concentrations and reduced NET cells (BON and QGP) proliferation in a dose-dependent manner. In addition, the in vitro assay did not detect any cytotoxicity of the empty EV in BON and QGP cells (Figure 3B).

The Western blotting analysis of proliferation and apoptosis proteins showed that 10 nM Ver-A treatment significantly reduced the expression of proliferation signaling protein AKT (protein kinase B) and completely eradicated the expression of protein cyclin D1, P21 and P27 on BON cells (shown as sample 3). 5 nM Ver-A treatment did not impact the expression of AKT but did reduce cyclin D1 and P21 expression and almost eradicated P27 expression (shown as sample 2). Different from BON cells, the 5 or 10 nM Ver-A did not change the expression of AKT while completely blocked the expression of cyclin D1, p21 and p27 in QGP cells. These results are consistent with literature reporting that Ver-A downregulated the expression of Akt signaling proteins in pancreatic adenocarcinoma and prostate cancer [55,56]. Unlike Ver-A, it is found that 10 or 20 nM DM1 did not change the expression of AKT, cyclin D1, p21 and p27 in BON and QGP cells. In addition to free drugs, we also tested the EV-delivered combined 5 nM Ver-A and 5 nM DM1. Surprisingly, Bon and QGP cells treated with 5 nM EV-Ver-A/DM1 (sample 6) exhibited the same Western blot pattern as that treated with 10 nM free Ver-A drug. This is probably attributed to the better biocompatibility of EV, which leads to more efficient drug delivery than free drugs. In addition, unlike EV-encapsulated drugs, the hydrophobic free drugs (such as Ver-A) were exposed to a hydrophilic cell culture environment during treatment with harsh pH and temperature. Free drugs are more prone to degradation (such as hydrolysis) than the EV-encapsulated drugs, which could compromise the potency of free drugs during treatment. Overall, these data indicated that Ver-A and DM1 were involved in the regulation of proliferation and apoptosis of NET cells although the anticancer mechanisms need further investigation.

### 3.4. Maximum Tolerated Dosage (MTD) and Minimal Toxicity

To investigate the MTD and potential systemic toxicity, we treated the nude mice with five doses of anti-SSTR2/CXCR4 EV-Ver-A/DM1: 0.31, 0.62, 1.24, 2.48, and 4.96 mg/kg. The bodyweight and behavior of the mice were monitored twice daily for a total of eight days. There were no overt changes in water intake, breathing and locomotion, and no obvious effects on body weight or overall survival for doses of 0.31–1.24 mg/kg (Figure 4A). However, the dose of 100 and 200 nM Ver-A and DM1 reduced body weight by ~10% during the first three days post-treatment, then the body weight started increasing from day 4. At the end of this study, mice were sacrificed to collect the important organs, such as brain, lung, heart, kidney and liver, for further toxicity investigation. As shown in H&E staining, none of these organs in the mice treated with 200 nM Ver-A and DM1 had obvious morphology change or necrosis as compared to PBS control group (Figure 4B). These results indicated that these five dosages of anti-SSTR2/CXCR4 mAbs-EV-Ver-A/DM1 tested in this study had no evident off-target effects in vivo. Considering the change of body weight at 2.48 and 4.96 mg/kg, we used the dose of 1.24 mg/kg in the in vivo anticancer efficacy study.

### 3.5. In Vivo Anti-NET Efficacy

The nude mice bearing BON-Luc/QGP xenografts were treated in a dosing interval of five days with PBS, SSTR2 mAb-EV-DM1, SSTR2 mAb-EV-Ver-A/DM1 and SSTR2/CXCR4 mAbs-EV-Ver-A/DM1 at dose of 1.24 mg/kg in 4 groups (*n* = 6). Figure 5A showed that the NET tumor growth was significantly attenuated in the SSTR2/CXCR4 dual-targeted EV-Ver-A/DM1 treatment group compared to PBS control group (*p* ≤ 0.005). The profiles of bodyweight revealed that there was no difference of bodyweight among all the four groups (Figure 5B), indicating the overall therapeutic safety and the lack of off-target toxicity during treatment. The wet weight measurement of terminal tumors confirmed the significant anti-NET efficacy of dual-targeted combined therapies (Figure 5C,D), which reduced tumor growth rate by 71%. The SSTR2 single-targeted combined therapies reduced the tumor growth rate by 39% and the single-targeted DM1 inhibited tumor growth rate only by 27%. These data indicated that our SSTR2/CXCR4 dual-targeted mAbs-EV is an effective drug delivery vehicle for highly potent reagents that are too toxic as conventional chemotherapies (Ver-A and DM1 in this study). Moreover, dual targeting had higher anticancer efficacy than single targeting, and targeted EV enabled the delivery of combined chemotherapies to effectively treat cancers with integrated anticancer mechanisms.

## 4. Conclusions

The EVs have great potential as drug delivery vehicles to treat cancers, but there are some technical challenges to apply EV-based drug delivery in clinics, such as cancer-specific targeting, toxicity or safety, pharmacokinetics or circulation stability, and cost effective biomanufacturing. The developed dual-targeted drug delivery of highly potent anticancer chemotherapies using EV could partially overcome the issues. Moreover, the developed mAbs-EV-drugs can be easily adapted to treat different cancers or tumors by tagging cancer surface receptors-specific antibodies. Despite the achievement so far, the targeting delivery platform of EV requires further development or evaluation before clinical application. For instance, the circulation stability of the constructed mAbs-EV-drugs, such as the stability of linker bridging mAbs and EVs and the stability of packed potent drugs, needs further optimization and full investigation. In addition to inhibiting tumor growth, the loading and delivering capability of combined chemotherapies, gene therapies and biotherapies needs further evaluation, aiming to eliminate tumor cells in vivo. Moreover, a pharmacokinetics study is highly desired to assess the circulation stability of the dual-targeted system-delivered therapies.

## Figures and Tables

**Figure 1 pharmaceutics-12-01079-f001:**
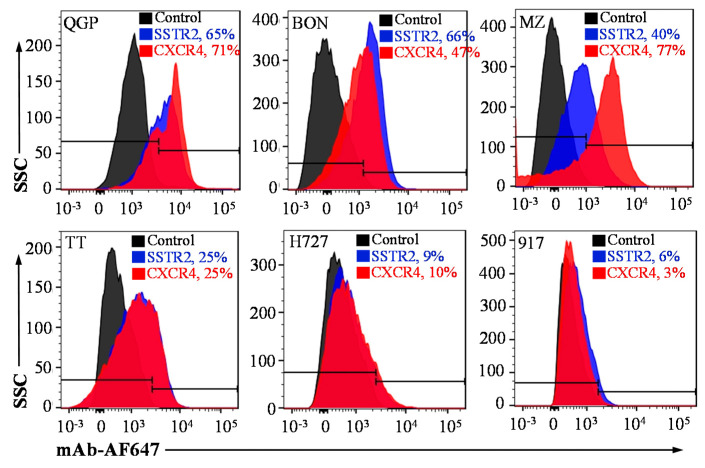
Evaluation of the expression of surface receptors anti-somatostatin receptor (SSTR2) and anti-C-X-C motif chemokine receptor 4 (CXCR4). Flow cytometry analysis of pancreatic neuroendocrine tumors (NETs) (QGP and BON), thyroid NETs (MZ and TT), and normal cells (917) stained with 1 μg of mAb-AF647/million cells on ice for 30 min.

**Figure 2 pharmaceutics-12-01079-f002:**
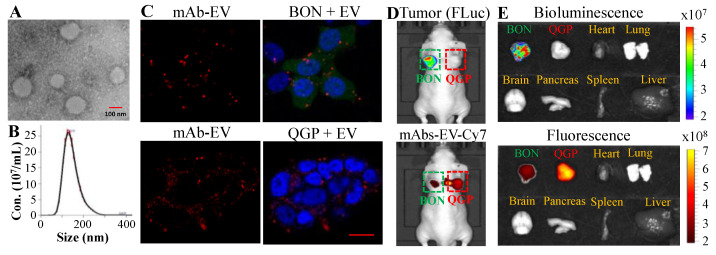
Evaluation of the NET-specific targeting of the SSTR2/CXCR4 dual-targeted monoclonal antibodies tagged extracellular vesicle (mAbs-EV). (**A**) Transmission electron microscope (TEM) image of mAbs-EV. (**B**) Size distribution analysis by NanoSight. (**C**) Live-cell confocal laser scanning microscopy (CLSM) dynamic imaging of QGP and BON cells mixed with mAbs-EV-Liss Rhod. Two or three-color CLSM: whole cell labeled with green fluorescent protein (GFP) (displayed as green), nucleus labeled with 4′,6-diamidino-2-phenylindole (DAPI) (displayed as blue), and mAbs-EV with Liss Rhod (displayed as red). Scale bar equals 10 µm. (**D**) In vivo live-animal In Vivo Imaging System (IVIS) imaging to analyze cancer specific targeting and biodistribution of mAbs-EV-AAV 24 h postinjection in NET xenograft mouse model (*n* = 4). (**E**) Ex vivo IVIS imaging of tumor and important organs 24 h postinjection.

**Figure 3 pharmaceutics-12-01079-f003:**
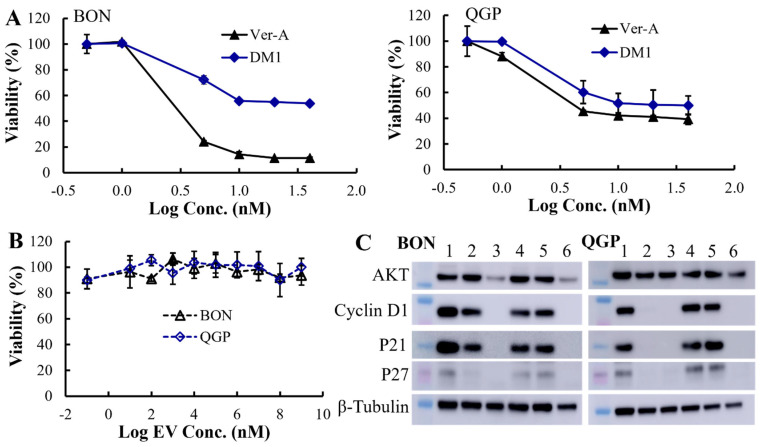
In vitro anticancer evaluation. (**A**) The anticancer cytotoxicity IC_50_ of free drug Ver-A (♦) and DM1 (▲) in BON and QGP cells (data represent mean ± SEM, *n* = 3). (**B**) No cytotoxicity of empty EV was detected. (**C**) Western blotting analysis of proliferation and apoptosis biomarkers in BON and QGP. 1: PBS control; 2: 5 nM Ver-A; 3: 10 nM Ver-A; 4: 10 nM DM1; 5: 20 nM DM1; and 6: 5 nM EV-Ver-A/DM1.

**Figure 4 pharmaceutics-12-01079-f004:**
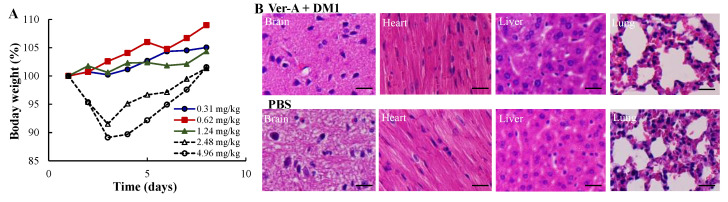
Maximum tolerated dose (MTD) and toxicity analysis of SSTR2/CXCR4 dual-targeted mAbs-EV-ver-A/DM1. (**A**) MTD to test the effect of five dosages including 0.31, 0.62, 1.24, 2.48, 4.96 mg/kg. *n* = 3. (**B**) Haematoxylin and eosin (H&E) staining of main organs, including brain, heart, liver and lung. Scale bar equals to 20 µm.

**Figure 5 pharmaceutics-12-01079-f005:**
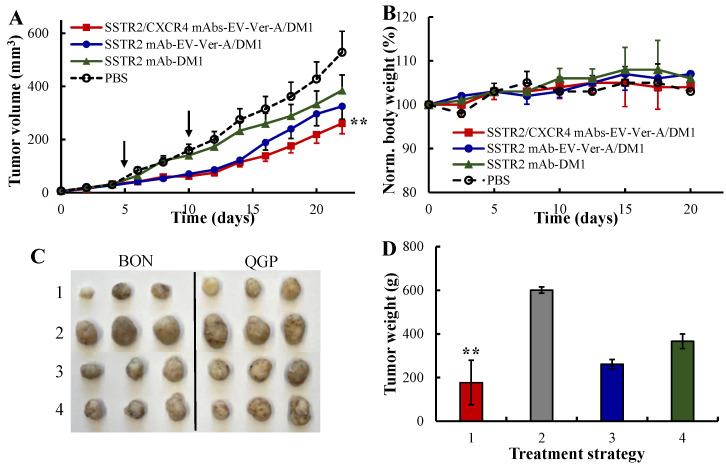
In vivo anticancer efficacy of dual-targeted Ver-A/DM1 in NET (BON-Luc and QGP) xenografted mouse model. (**A**) Tumor volume changes in NET xenografts followed the treatment (data represent mean ± SEM, *n* = 6). 3 × 10^6^ mycoplasma-free BON-Luc or QGP cells were subcutaneously injected into nude mice. PBS (**○**), single-targeted anti-SSTR2 mAb-EV-DM1 (▲), single-targeted anti-SSTR2 mAb-EV-Ver-A/DM1 (●), and dual-targeted anti-SSTR2/CXCR4 mAbs-EV-Ver-A/DM1 (■) on days 5 and 10. Tumor volume was measured with calipers, and calculated as ellipsoid. ** *p* < 0.005 vs. PBS using ANOVA followed by Dunnett’s *t*-test. Black arrow indicating the injection date of the targeting delivered Ver-A/DM1 or DM1 (1.24 mg/kg). (**B**) The normalized body weight. (**C**) Representative tumors harvested. (**D**) The total wet weight of the harvested NET (BON and QGP) xenograft tumors. ** *p* < 0.005 vs. PBS using ANOVA followed by Dunnett’s *t*-test.

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
