# Peer review of "Dual-Targeted Extracellular Vesicles to Facilitate Combined Therapies for Neuroendocrine Cancer Treatment"

_pharmaceutics, 2020, doi:10.3390/pharmaceutics12111079_

Round 1

Reviewer 1 Report

The work by Si et al. focus on the study of extracellular vesicles-mediated antibody targeting in neuroendocrine (NE) cancers. This study, based on the technology tagging monoclonal antibodies on extracellular vesicles aimed to develop and validate a dual-targeted EVs to deliver chemotherapeutic drugs to specific targets in NET. SSTR2 and CXCR4 are found here to be optimal targets for the dual-targeted treatment for NETs, overcoming the limitation of treatment of heterogenous cancers. Verrucarin A (Ver-A) and mertansine (DM1) are used as drugs that can be loaded on extracellular vesicle and kill cancer cells.

The authors used in vitro and in vivo experiments to evaluate the effect of the engineered vesicles bearing mAb anti-SSTR2/CXCR4 loaded with Ver-A/DM1 drugs and brought evidences of tumor growth inhibition.

While the aim is of sound interest and the manuscript is clearly written, conclusions are brief and limitations of the study are not acknowledged.

Specific:

  • It should be clear which population of extracellular vesicle the study refers to, are all ‘exosomes’? To date it is clear that there are several populations of extracellular vesicles, that given the different purification methods, can cause variability in the overall conclusions. Unless it is clearly exosomes that are purified (of endosomal origin), it would be more correct to use the general term of extracellular vesicle.
  • Figure 3/4: a basic control only with untagged exosomes or empty exosomes is missing

Minor:

  • Line 76: correct ‘exesomes’ to ‘exosomes’
  • Fig 5 Which are the cells used here? Treatments 1-4?

Author Response

Manuscript: pharmaceutics-966604, Dual-targeted exosomes to facilitate combined therapies for neuroendocrine cancer treatment

We thank the reviewer 1 for the insightful comments, which significantly improve the quality of this article. We have revised the manuscript following the reviewer’s suggestions and answered all questions. These changes were highlighted in the manuscript and also described below.

Comments and Suggestions for Authors

The work by Si et al. focus on the study of extracellular vesicles-mediated antibody targeting in neuroendocrine (NE) cancers. This study, based on the technology tagging monoclonal antibodies on extracellular vesicles aimed to develop and validate a dual-targeted EVs to deliver chemotherapeutic drugs to specific targets in NET. SSTR2 and CXCR4 are found here to be optimal targets for the dual-targeted treatment for NETs, overcoming the limitation of treatment of heterogenous cancers. Verrucarin A (Ver-A) and mertansine (DM1) are used as drugs that can be loaded on extracellular vesicle and kill cancer cells.

The authors used in vitro and in vivo experiments to evaluate the effect of the engineered vesicles bearing mAb anti-SSTR2/CXCR4 loaded with Ver-A/DM1 drugs and brought evidences of tumor growth inhibition.

While the aim is of sound interest and the manuscript is clearly written, conclusions are brief and limitations of the study are not acknowledged.

Response to Reviewer 1 Comment No. 1: We thank the reviewer comment. The following description of the limitations of this study was added into “Conclusion”.

  1. Conclusion, Lines 346-354: “The circulation stability of the constructed mAbs-Exo-drugs, such as the stability of linker bridging the mAbs and exosomes and the stability of the packed potent drugs, is very important to apply this targeted delivery system in future clinics. In addition, the tumor growth was significantly inhibited by the tested dosage and treatment strategy of anti-SSTR2/CXCR4 mAbs-Exo-VerA/DM1, but the tumor cells were not eliminated in vivo. In future, we will perform a pharmacokinetics study to evaluate the in vivo stability of the dual-targeted system, test its targeting delivery capability of more chemotherapies and optimize the drug loading strategy, and fully investigate the anti-cancer efficacy of this system by further optimizing the dosage and treatment schedule of different combined therapies.”

Specific:

  • It should be clear which population of extracellular vesicle the study refers to, are all ‘exosomes’? To date it is clear that there are several populations of extracellular vesicles, that given the different purification methods, can cause variability in the overall conclusions. Unless it is clearly exosomes that are purified (of endosomal origin), it would be more correct to use the general term of extracellular vesicle.

Response to Reviewer 1 Comment No. 2: We agree with the reviewer that there are different populations of extracellular vesicles. We have characterized the isolated exosomes using NanoSight and TEM in this study, confirmed the endosomal origin of our exosomes using Western blotting to detect the markers of GAPDH, HSP70, CD9, CD63 and CD81, and also confirmed the exosomes purify with Western showing the lack of contamination of ER (CANX), Golgi (GM130), mitochondria (CYTO C) and nuclear (HISTONES) in our recent publication (Si, et al, Biotechnol. J., 2020). To clarify, we modified the manuscript as below.    

  1. Materials and Methods. Lines 126-129: “The generated mAbs-Exo were titrated with NanoSight (Malvern Panalytical, Malvern, UK) with 5 captures in 5 minutes per sample using SCMOS camera at room temperature, pump speed of 25 and detection threshold of 5. The morphology of exosomes was confirmed with a Tecnai Transmission electron microscope (TEM, FEI, Hillsboro, OR).

Figure 2. Lines 246-249:

Figure 2. Evaluation of the NET-specific targeting of the SSTR2/CXCR4 dual-targeted mAbs-Exo. (A) Transmission electron microscope (TEM) image of mAbs-Exo. (B) Size distribution analysis by NanoSight.

  1. Results and Discussion. Lines 236-241: “The homogeneity, quality and size distribution of mAbs-Exo were characterized. The transmission electron microscope (TEM) image showed that the targeted exosomes were physically homogenous (Figure 2A). The nanoparticle tracking analysis using NanoSight showed that the mAbs-Exo had a mean size of 145±39 nm with size distribution of 102-194 nm (Figure 2B). Moreover, our previous study confirmed the endosomal origin of the generated exosomes and high purity without contamination of ER, Golgi, mitochondria and nuclear [24].”

As reported in our previous publication (Si, et al, Biotechnol. J., 2020), Western blotting analysis confirmed the following markers in exosomes generated and purified using our developed biomanufacturing, glyceraldehyde 3-phosphate dehydrogenase (GAPDH), heat shock protein 70 (HSP70), and surface receptors of tetraspanins including CD9, CD63 and CD81. Also, we did not detect the presence of cellular debris contaminants by analyzing the CANX (ER marker), GM130 (Golgi marker), CYTO C (mitochondrial marker) and HISTONES (nuclear marker), which confirmed the high purity of our endosomal origin exosomes (Figure below).

  • Figure 3/4: a basic control only with untagged exosomes or empty exosomes is missing

Response to Reviewer 1 Comment No. 3: As suggested by the reviewer, we added the in vitro anti-cancer cytotoxicity results using the empty exosomes into Figure 3. Since empty exosomes vehicle has no cytotoxicity, we did not test it in in vivo study in Figure 4. 

  1. Results and Discussion. Lines 280-281: “In addition, the in vitro assay did not detect any cytotoxicity of the empty exosomes in BON and QGP cells (Figure 3B).”

Figure 3. Lines 268-269:

Figure 3. In vitro anti-cancer evaluation. (A) The anti-cancer cytotoxicity IC50 of free drug Ver-A (♦) and DM1 (â–²) in BON and QGP cells (data represent mean ± SEM, n = 3). (B) No cytotoxicity of empty exosomes vehicle was detected. (C) Western blotting analysis of proliferation and apoptosis biomarkers in BON and QGP. 1: PBS control; 2: 5 nM Ver-A; 3: 10 nM Ver-A; 4: 10 nM DM1; 5: 20 nM DM1; and 6: 5 nM Exo-Ver-A/DM1.

Minor:

Line 76: correct ‘exesomes’ to ‘exosomes’

Response to Reviewer 1 Comment No. 4: We thank the reviewer’s advice. The typo was corrected in Line 76 (Line 78 in the revised manuscript).

Reviewer 1 Comment 5. Fig 5 Which are the cells used here? Treatments 1-4?

Response to Reviewer 1 Comment No. 5: We modified the Figure 5D legend Lines 339-340 as “The total wet weight of the harvested NET (BON and QGP) xenograft tumors.” 

Reviewer 2 Report

The authors present an interesting approach in the direction of targeted use of exosomes in therapy but if general premises are fascinating, the presented results miss the details of the crucial reagents: the exosomes. In fact, not a single Western blotting validating the correct purification methods are presented blotting is presented, or a FACS analysis, or a nanosizer or SEM/TEM analysis. Nothing. Also in the "Materials and Methods" section, the protocol of exosomes purification is more or less completely missed. Please improve the underlined gaps. In order to the introductive part, exosomes are not only used for therapy but are strongly used for cancer progression monitoring (see PMID: 32759810, PMID: 29029605, and others). I look forward to receiving the revised version of the paper and I hope that my comments can be useful.

Author Response

Manuscript: pharmaceutics-966604, Dual-targeted exosomes to facilitate combined therapies for neuroendocrine cancer treatment

We thank the reviewer 2 for the insightful comments, which significantly improve the quality of this article. We have revised the manuscript following the reviewer’s suggestions and answered all questions. These changes were highlighted in the manuscript and also described below.

Comments and Suggestions for Authors

The authors present an interesting approach in the direction of targeted use of exosomes in therapy but if general premises are fascinating, the presented results miss the details of the crucial reagents: the exosomes. In fact, not a single Western blotting validating the correct purification methods are presented blotting is presented, or a FACS analysis, or a nanosizer or SEM/TEM analysis. Nothing.

Response to Reviewer 2 Comment No. 1: We thank the reviewer’s suggestions. We have characterized the isolated exosomes using NanoSight and TEM in this study and also characterized the stirred-tank bioreactor-based exosomes using Western blotting by detecting multiple markers previously as reported in our publication (Si, et al, Biotechnol. J., 2020). We modified the manuscript as below.    

  1. Materials and Methods. Lines 126-129: “The generated mAbs-Exo were titrated with NanoSight (Malvern Panalytical, Malvern, UK) with 5 captures in 5 minutes per sample using SCMOS camera at room temperature, pump speed of 25 and detection threshold of 5. The morphology of exosomes was confirmed with a Tecnai Transmission electron microscope (TEM, FEI, Hillsboro, OR).

Figure 2. Lines 246-249:

Figure 2. Evaluation of the NET-specific targeting of the SSTR2/CXCR4 dual-targeted mAbs-Exo. (A) Transmission electron microscope (TEM) image of mAbs-Exo. (B) Size distribution analysis by NanoSight.

  1. Results and Discussion. Lines 236-241: “The homogeneity, quality and size distribution of mAbs-Exo were characterized. The transmission electron microscope (TEM) image showed that the targeted exosomes were physically homogenous (Figure 2A). The nanoparticle tracking analysis using NanoSight showed that the mAbs-Exo had a mean size of 145±39 nm with size distribution of 102-194 nm (Figure 2B). Moreover, our previous study confirmed the endosomal origin of the generated exosomes and high purity without contamination of ER, Golgi, mitochondria and nuclear [24].”

As reported in our previous publication (Si, et al, Biotechnol. J., 2020), Western blotting analysis confirmed the following markers in exosomes generated and purified using our developed biomanufacturing, glyceraldehyde 3-phosphate dehydrogenase (GAPDH), heat shock protein 70 (HSP70), and surface receptors of tetraspanins including CD9, CD63 and CD81. Also, we did not detect the presence of cellular debris contaminants by analyzing the CANX (ER marker), GM130 (Golgi marker), CYTO C (mitochondrial marker) and HISTONES (nuclear marker), which confirmed the high purity of our endosomal origin exosomes (Figure below).

Also in the "Materials and Methods" section, the protocol of exosomes purification is more or less completely missed. Please improve the underlined gaps.

Response to Reviewer 2 Comment No. 2: As suggested by the reviewer, we added more details in exosomes purification in the revised manuscript.

  1. Materials and Methods. Lines 111-119: “When HEK293F cell viability dropped to 80-90% in bioreactor, the culture broth was collected and centrifuged at 4,000 g and 4oC for 1 hour. Then the collected spent medium was filtered to remove cell debris and treated with proteinase K to remove secreted host cell proteins. The exosomes were purified, concentrated and washed using Vivaspin 300 kDa MWCO concentrators. Finally, exosomes were further purified and washed using Vivaspin turbo 100 kDa MWCO concentrators. In this study, the fine polishing using Protein A chromatography was not performed. The purified exosomes were used for surface tagging and drug packing or supplemented with 25 mM trehalose for long-term storage at -80oC.”

In order to the introductive part, exosomes are not only used for therapy but are strongly used for cancer progression monitoring (see PMID: 32759810, PMID: 29029605, and others).

Response to Reviewer 2 Comment No. 3: As suggested by the reviewer, we modified the Introduction as below.

  1. Lines 40-42: “Exosomes, extracellular nanovesicles, play important roles in cell-cell communication via transporting miRNA, siRNA, DNA, and proteins in vivo, and also have great potential to deliver drug for cancer treatment and to monitor cancer progression [18-23].”

I look forward to receiving the revised version of the paper and I hope that my comments can be useful.

Response to Reviewer 2 Comment No. 4: We addressed all the reviewers’ comments, revised the manuscript and re-submitted.

Round 2

Reviewer 1 Report

I appreciate the authors' responses to the comments, however the quality of the manuscript did not improve.

Comment 1: conclusions must be improved, in terms of content and form.

Comment 2: different populations of extracellular vesicles are present. The markers you refer to (and show in another work) are found in all different kind of vesicles and do not per se represent a distinctive marker (except for CD63), moreover from the western blot presented in the rebuttal letter, it is not clear what is shown in the figure. Exosome lysates? The total cell lysates to control for the non exosomal markers, such as GM130 are absent.

Exosome  term refers to smaller vesicles than 150nm, to be sure exosomes are purified more experimental data have to be shown. I would then suggest to use the term EV.

Author Response

Manuscript: pharmaceutics-966604, Dual-targeted extracellular vesicles to facilitate combined therapies for neuroendocrine cancer treatment

We thank the reviewer 1 for the insightful comments and have revised the manuscript following the reviewer’s suggestions. These changes were highlighted in the manuscript and also described below.

Comments and Suggestions for Authors

I appreciate the authors' responses to the comments, however the quality of the manuscript did not improve.

Comment 1: conclusions must be improved, in terms of content and form.

Response to Reviewer 1 Comment No. 1: As reviewer suggested, we did major revision in the “Conclusions”, Lines 345-357.

The EVs have great potential as drug delivery vehicles to treat cancers, but there are some technical challenges to apply EVs-based drug delivery in clinics, such as the cancer-specific targeting, toxicity or safety, pharmacokinetics or circulation stability, and cost effective biomanufacturing. The developed dual-targeted drug delivery of highly potent anti-cancer chemotherapies using EV could partially overcome the issues. Moreover, the developed mAbs-EV-drugs can be easily adapted to treat different cancers or tumors by tagging cancer surface receptors-specific antibodies. Despite the achievement so far, the targeting delivery platform of EV requires further development or evaluation before clinical application. For instance, the circulation stability of the constructed mAbs-EV-drugs, such as the stability of linker bridging mAbs and EVs and the stability of packed potent drugs, needs further optimization and full investigation. In addition to inhibiting tumor growth, the loading and delivering capability of combined chemotherapies, gene therapies and biotherapies needs further evaluation, aiming to eliminate tumor cells in vivo. Moreover, a pharmacokinetics study is highly desired to assess the circulation stability of the dual-targeted system-delivered therapies.

Comment 2: different populations of extracellular vesicles are present. The markers you refer to (and show in another work) are found in all different kind of vesicles and do not per se represent a distinctive marker (except for CD63), moreover from the western blot presented in the rebuttal letter, it is not clear what is shown in the figure. Exosome lysates? The total cell lysates to control for the non exosomal markers, such as GM130 are absent.

Exosome term refers to smaller vesicles than 150nm, to be sure exosomes are purified more experimental data have to be shown. I would then suggest to use the term EV.

Response to Reviewer 1 Comment No. 2: As reviewer suggested, we changed “exosomes (Exo)” to “extracellular vehicle (EV)” in the manuscript in Lines 2, 16-17, 19-20, 24-28, 42, 45-46, 49, 67, 78, 80-8, 83-84, 106-108, 110-112, 116-117, 119-120, 125-127, 129, 131-132, 137-139, 146, 149-150, 152, 164, 1667-168, 177-179, 185-187, 189, 223, 227, 236, 237-240, 242-243, 246, 248-252, 254, 256, 258-260, 262-263, 266-267, 269, 272-273, 283, 294, 301, 310, 314, 319, 321, 328, 331, 333, 337-339, 346, 349-350, 352,  

Reviewer 2 Report

Thanks to the author's efforts, the article is now suitable for publication.

Thanks for the opportunity to revise this nice work,

Author Response

Manuscript: pharmaceutics-966604, Dual-targeted extracellular vesicles to facilitate combined therapies for neuroendocrine cancer treatment

Comments and Suggestions for Authors

Comment: Thanks to the author's efforts, the article is now suitable for publication.

Response to Reviewer 2 Comment: We thank the reviewer 2 for all previous insightful comments, which significantly improved the quality of this article.

Round 3

Reviewer 1 Report

Despite the efforts of the authors, I am not yet well convinced the criticism has been elaborated and that manuscript deserves publication in this journal.